# Prognostic Value of Carcinoembryonic Antigen (CEA) and Carbohydrate Antigen 19-9 (CA 19-9) in Gallbladder Cancer; 65 IU/mL of CA 19-9 Is the New Cut-Off Value for Prognosis

**DOI:** 10.3390/cancers13051089

**Published:** 2021-03-04

**Authors:** Myongjin Kim, Hongbeom Kim, Youngmin Han, Heeju Sohn, Jae Seung Kang, Wooil Kwon, Jin-Young Jang

**Affiliations:** Department of Surgery, Seoul National University Hospital, Seoul 03080, Korea; 5C322@snuh.org (M.K.); 65922@snuh.org (H.K.); views@snu.ac.kr (Y.H.); 74662@snuh.org (H.S.); 83307@snuh.org (J.S.K.); willdoc@snu.ac.kr (W.K.)

**Keywords:** gallbladder neoplasm/analysis, gallbladder neoplasm/surgery, tumor marker, cut off value, overall survival

## Abstract

**Simple Summary:**

Gallbladder cancer (GBC) is the fifth most common cancer of the digestive tract, and preoperative tumor markers for GBC have been studied as a less invasive way to detect the presence of the cancer. CEA and CA 19-9 have been most commonly used for detecting GBC clinically, and various cut-off values were suggested to satisfy this purpose, but there has still been a lack of proper values of these tumor markers to predict the prognosis of GBC. We have aimed to suggest appropriate cut-off values that could help to anticipate prognosis in the preoperative period. Data from carefully selected 539 patients were used in our study, the new cut-off value, 65 IU/mL for CA 19-9 was derived through an up-to-date statistical method. By using this cut-off value, clinicians could get the important reference in the establishment of the strategy of treatment, and the researches about this topic could become more vigorous.

**Abstract:**

Due to the lack of appropriate tumor markers with optimal cut-off values to predict the prognosis of gallbladder cancer (GBC), this study aimed to demonstrate the relationship between prognosis and the levels of carcinoembryonic antigen (CEA) and carbohydrate antigen 19-9 (CA 19-9), and to determine optimal thresholds. In total, 539 patients diagnosed with GBC were examined. The relationship between tumor marker levels and overall survival (OS) was analyzed. The C-tree method was used to suggest tumor marker thresholds, and multivariate analysis was conducted to identify prognostic factors for overall survival. The mean age of the patients was 65.3 years, and the 5-year overall survival rate in all patients was 68.9%. Following the C-tree method, the optimal cut-off value was set at 5 IU/mL for CEA and at 65 IU/mL for CA 19-9. Multivariate analysis revealed that age, CA 19-9 level, operative method, T stage, and N stage were significant prognostic factors for OS. Consequently, CA 19-9 had a stronger association with prognosis than CEA, and 65 IU/mL for CA 19-9 may be suggestive in evaluating the prognosis of GBC. Moreover, it could be an effective indicator for determining the surgical extent necessary and the need for adjuvant treatment.

## 1. Introduction

Tumor markers can be acquired in a less invasive and harmful manner than other methods like radiologic or endoscopic methods [1]. Several tumor markers are associated not only with diagnosis but also with prognosis and can therefore be useful in determining treatment strategies. Recently, efforts have been made to find appropriate tumor markers with sufficient power to predict the prognosis of various cancers [2,3]. 

Gallbladder cancer (GBC) is the most common malignancy of the biliary tract and the fifth most common malignant tumor of the digestive tract, and many studies have investigated appropriate tumor markers of GBC. For instance, elevation of carcinoembryonic antigen (CEA) and carbohydrate antigen 19-9 (CA 19-9) increase the suspicion of such a diagnosis [4,5]. Several studies have focused on the relationship between diagnosis and the level of tumor markers. In these studies, the cut-off values varied, ranging from 4 IU/mL for CEA and 10 to 20 IU/mL for CA 19-9 [6,7]. These cut-off values can be used in detecting the presence of malignancy.

Furthermore, our study focused on predicting the prognosis of GBC patients using these tumor markers because preoperatively, predicted prognosis is very important to determine the future treatment strategy for GBC. Surgical treatments for GBC such as simple cholecystectomy, extended cholecystectomy can be performed through an open laparotomy, laparoscopically, or robotically. In addition, various kinds of adjuvant treatment such as chemotherapy, radiotherapy, or chemoradiation could be considered according to the stage of the cancer. Pathologic staging is obviously the most powerful factor that can help determine treatment options and predict prognosis [8]. However, this factor can only be acquired postoperatively, which suggests the need for a method that can predict prognosis prior to surgery. CEA, CA 19-9, and tumor markers for gastrointestinal and pancreatobiliary malignancies have been the most commonly investigated tumor markers [1]. However, optimal cut-off values for predicting the prognosis of GBC, and the method of determining these values vary from study to study. 

Therefore, the aim of this study was to demonstrate the relationship between the prognosis of GBC and the level of tumor markers such as CEA and CA 19-9, and to determine the optimal cut-off values for predicting prognosis. 

## 2. Materials and Methods

### 2.1. Patients and Data Collection

A total of 539 patients who underwent curative surgery for GBC at Seoul National University Hospital between January 2000 and July 2019 were retrospectively identified. The patients were limited to those who underwent R0 resection with documented preoperative tumor markers. The pathology of all patients was confirmed to be adenocarcinoma (AC). Patients who died within 30 days postoperatively, received R1/R2 resection, had distant metastasis (M1), and had insufficient medical data were excluded. Blood samples for CEA and CA 19-9 levels were taken within 2 weeks prior to surgery.

All data including age, sex, total bilirubin, preoperative CEA and CA 19-9 levels, type of operation, histologic differentiation, T stage, N stage, pathologic type of tumor, complications, adjuvant treatment, and recurrence were analyzed for this study. 

This study was approved by the Institutional Review Board (IRB) (SNUH-1812-002-989).

### 2.2. Statistical Analysis

Continuous variables were expressed as mean and standard variation, and were compared using the Students t-test. Tumor markers were expressed as median values and as interquartile ranges (IQRs). Additionally, all categorical variables were described as numbers and percentages. Categorical variables were compared using the Pearson’s *X^2^* test. Overall survival (OS) was defined as the period from the time of resection to death or last follow-up. Disease-free survival (DFS) was calculated as the period from the time of recurrence or death. The survival rate was determined using the Kaplan–Meier method.

To find the optimal cut-off values that could clearly stratify survival outcomes, the conditional inference tree (C-tree) method, which uses recursive partitioning of dependent variables based on the value of correlation, was used. After obtaining the optimal cut-off values, they were compared to the clinical upper normal limit of 5 IU/mL for CEA and 37 IU/mL for CA 19-9.

Multivariate analyses were performed using the Cox proportional hazard model to identify significant factors that could affect survival rate. All statistical analyses were performed using SPSS version 25.0 (IBM SPSS Statistics, IBM Corp. Armonk, NY, USA) and R software (version 3.1.2; R Foundation for Statistical Computing). Statistical significance was set at *p* < 0.05.

## 3. Results

### 3.1. Demographics 

From January 2000 to July 2019, 744 patients underwent surgery at Seoul National University Hospital. Thirty-one patients were excluded because of insufficient medical data. Three patients who died within 30 days postoperatively, 60 patients who had received R1/R2 resection, and 111 patients with M1 stage were also excluded. Finally, 539 patients were enrolled (Figure 1).

The mean age of the patients was 65.3 years, with 255 (47.3%) men and 284 (52.7%) women. One hundred and seventy-one patients underwent simple cholecystectomy and 306 patients underwent extended cholecystectomy. Additionally, 26.5% were of stage T1, 54.9% were of stage T2, 17.4% were of stage T3, and 1.1% were of stage T4. There were 144 (26.7%) patients with lymph node metastasis, and 395 (73.3%) patients did not have lymph node metastasis. Forty-five patients (8.3%) experienced complications, including intra-abdominal fluid collection and wound problems after the operation. Eighty-five (15.8%) and 76 (14.1%) patients received chemotherapy and radiotherapy in the postoperative period, respectively (Table 1).

### 3.2. Tumor Marker Distribution

In our study population, the median value of CEA was 1.80 IU/mL with IQR, 1.20–2.70. The median value of CA 19-9 was 13.2 IU/mL with IQR, 6.00–32.78. The distribution of preoperative CEA and CA19-9 levels is shown in Appendix A. Median values of CEA were 1.80 IU/mL in T1 stage, 1.70 IU/mL in T2 stage, 2.10 IU/mL in T3 stage, and 2.50 IU/mL in T4 stage (*p* = 0.016). Additionally, median values of CA 19-9 were 9.00 IU/mL in T1 stage, 12.00 IU/mL in T2 stage, 39.55 IU/mL in T3 stage, and 23.00 IU/mL in T4 stage (*p* = 0.894). A similar pattern of difference was observed in the N stages. Log transformation was performed on the tumor markers because they were not normally distributed (Appendix A). There was a tendency for tumor markers to be elevated in association with elevated T and N stages, even in log-transformed equations, and the result was statistically significant from the *p*-values of each distribution. 

### 3.3. Survival According to the Tumor Marker and Optimal Cut Off Value 

The 5-year overall survival rate (5-year OS) of all patients was 68.9%. The 5-year OS was 90.8% for T1, 68.3% for T2, 39.0% for T3, and 20.8% for T4. The 5-year disease-free survival rate (5-year DFS) in all patients was 65.8%. When 5-year DFS was analyzed by T staging, it was found to be 89.7% for T1, 64.1% for T2, 34.3% for T3, and 0.0% for T4. 

The optimal cut-off values of 5 IU/mL and 65 IU/mL were derived from the C-tree method for CEA and CA 19-9, respectively (Figure 2). The 5-year OS of the patients who had a CEA level below 5 IU/mL was 72.1%, and was 24.2% (*p* < 0.001) in patients with CEA levels above 5 IU/mL. The 5-year OS of patients with a CA 19-9 level below 37 IU/mL was 77.2%, as compared to 37.8% in those with CA 19-9 levels above 37 IU/mL. The 5-year OS of patients with a CA 19-9 level below 65 IU/mL was 76.8% whereas that of the group with CA 19-9 level above 65 IU/mL was 24.0% (*p* < 0.001) (Figure 3). 

In our study, the rate of the recurrence associated with the cut-off value for predicting 5-year OS in patients with CA 19-9 levels above 65 IU/mL was 51.1%, as compared to 24.3%, that of total patients. Additionally, recurrence was shown to be 28.6% for stage I, 47.4% for stage II in patients with CA 19-9 levels above 65 IU/mL.

There was a difference in 5-year DFS when the total cohort was divided into two groups based on 65 IU/mL, it was derived that 5-year DFS of the patients who had a CA 19-9 level below 65 IU/mL was 73.7%, and was 23.9% in patients with CA 19-9 levels above 65 IU/mL.

Additionally, when a statistical analysis was performed with 65 IU/mL of CA 19-9, there was a significant difference in the proportion of T1, T2, and T3, when compared to setting the cut-off value at 37 IU/mL (Table 1). 

### 3.4. Prognostic Factors for Overall Survival 

Univariate analysis of OS indicated that age, operation type, T-stage, N-stage, complication, chemotherapy, preoperative CEA and CA 19-9 levels, and radiation therapy were statistically significant. However, multivariate analysis showed that preoperative CEA level > 5 IU/mL (HR: 1.613, *p* = 0.034), CA 19-9 > 65 IU/mL (HR: 2.557, *p* < 0.001), operation method (*p* = 0.004), T-stage (*p* = 0.007), and N-stage (*p* < 0.001) were statistically significant on multivariate analysis. CA 19-9 had a stronger connection with prognosis in comparison to CEA (HR: 1.613, CI: 1.037–2.510) (Table 2).

The value of 65 IU/mL of CA 19-9 was compared with 37 IU/mL, which is commonly used in the clinical field as a normal upper limit. This analysis showed that 65 IU/mL (HR: 2.557, CI: 1.763–3.710) was more significant in predicting overall survival than 37 IU/mL (HR: 1.876, CI: 1.331–2.644).

## 4. Discussion

CA 19-9 has been widely used as an indicator in the diagnosis and prognosis of GBC [4,5,7]. Although there have been several studies regarding the ideal cut-off value of CEA and CA 19-9 for predicting the prognosis of GBC, optimal cut-off values for CEA and CA 19-9 have not yet been established. Thus, 5 IU/mL for CEA and 65 IU/mL for CA 19-9 were our novel cut-off value derived from our statistical analysis, the C-tree method. The cut-off value of CEA corresponds to the current clinical upper normal limit, but a new cut-off value has been presented for CA 19-9. If it is shown that someone has high levels of the tumor marker, further evaluation could be considered for precise staging. In addition, if someone has a high level of the tumor marker in T1b stage, which is currently controversial in terms of strategy of treatment, lymph node sampling could be done for determination of extended cholecystectomy or accurate staging. Moreover, aggressive adjuvant therapy was carried out in the early staged group with a high level of the tumor markers.

There have been already several studies that have examined the optimal cut-off values of the tumor markers for GBC, but these studies had limitations that could not produce a definite reliability. First of all, some studies used normal upper limit of the tumor markers as cut-off values without any statistical method that could demonstrate the relevance of their cut-off values [9,10,11]. Secondly, some studies suggested new cut-off values which is not the clinical normal upper limit of the tumor markers, but it was not derived from the statistical way [6,7,12,13]. Other studies presented an insufficient number of experimental groups, lowering its statistical significance. [6,7,12,13]. A few studies were performed excessively on various kinds of diseases other than GBC [7,13], or were not carried out by a single center [6]. 

Other than CEA and CA 19-9, CA 242, that is known as specific diagnostic barometer for malignant biliary disease, was also examined as a prognostic indicator for GBC in this study, but it was revealed that this factor was not statistically significant to be used in predicting the prognosis of GBC (HR: 0.386, CI: 0.972–1.015, *p* = 0.535) [7]. Additionally, other tumor markers including AFP, CA 72-4 were examined by study of Liska V et al., but their statistical significance was not validated. [13].

The attempt to combine different tumor markers was not limited to only CEA and CA 19-9, Wei et al. utilized CA 19-9 and fibrinogen in 154 patients with GBC. The cut-off values were set as 25.45 IU/mL for CA 19-9 and 3.47 g/L for fibrinogen by using a receiver operating characteristic (ROC) curve. As a result, it was discovered that elevated CA 19-9 and fibrinogen levels indicated a worse prognosis [14]. Fibrinogen is a kind of acute-phase reactant that is synthesized in the liver and secreted into the circulation. In addition, it is known that the levels of this factor increase in response to most forms of tissue injury, infection, or inflammation. Furthermore, it was reported that fibrinogen promotes lymphatic and hematogenous metastases in a study which was based on mice. Based on this fact, Wei et al suggested the combination of fibrinogen with CA 19-9 as an indicator of prognosis for GBC. However, we thought CEA and CA 19-9 had more significant prognostic values for GBC than other factors. Therefore, our study focused on demonstrating their usefulness. 

In this study, we initially investigated both CEA and CA 19-9 levels, but the proportion of patients with increased CEA was only 7.3%. Therefore, CEA might not provide enough statistical significance in combination with CA 19-9; therefore, we did not investigate the combination of both these markers.

Many studies have investigated factors that can influence the prognosis of GBC. Positron emission tomography (PET) is an effective way to preoperatively obtain radiologic images that can provide information regarding the prognosis of GBC. Moradi et al. reported that high uptake of *fluorodeoxyglucose (*FDG) by glucose transporter1 (GLUT1) in PET could indicate a poor prognosis [15]. However, PET scans can present an economic burden for patients when compared to serological tumor markers, and can be a clinical drawback when applied to all GBC patients. A positive mucin 4 (MUC4) immunohistochemical method is associated with poor prognosis in bile duct carcinoma. Lee et al. reported that patients who had positive findings with MUC4 had significantly worse survival rates than those with negative findings (*p* = 0.048) [16]. These tests are very practical methods and provide useful information, but they can only be acquired after surgery, thus limiting their use. A serologic CA 19-9 study may stand out from the aforementioned methods due to its facility in collection and its inexpensiveness. Furthermore, it can be collected preoperatively, which may prove useful in terms of treatment options.

Whether adjuvant therapy has been performed can also be used as a prognostic factor. Mojica et al. reported improved survival in patients who received adjuvant radiation therapy for locally advanced GBC or GBC with regional disease in their study. The group that received adjuvant radiation therapy had a median survival of 14 months, which was significantly better than the median survival of 8 months in patients who did not receive adjuvant radiation therapy (*p* < 0.0001) [17]. Another study reported that chemoradiation prolongs survival. In this study, if the prognosis was limited to the T2N1M0 and T3N1M0 stages, DFS was higher in the groups that received adjuvant chemoradiation therapy than in the opposite groups in both stages [18]. 

However, other studies have been suspicious of the effect of these adjuvant therapies. Douglas et al. reported that there was no significant difference in DFS between the surgery-only group and adjuvant therapy group (*p* = 0.40) in their research [19]. Kalyan et al. conducted their own study to verify the efficacy of adjuvant therapy in GBC with 4775 patients and concluded that the survival benefit was uncertain in the group that received adjuvant chemotherapy [20].

We found that adjuvant therapy was an insignificant factor in the prognosis of GBC in our study. This may due to the fact that the number of patients who received chemotherapy or radiotherapy were small, being only 15.8% and 14.1%, respectively. In detail, the proportion of patients who received chemotherapy was 1.4% for T1, 15.5% for T2, 37.2% for T3, and 33.3% for T4. Additionally, the proportion of patients who received radiotherapy was 2.1% for T1, 13.5% for T2, 33.0% for T3, and 33.3% for T4. In detail, it was revealed that there was no benefit from adjuvant chemotherapy in the 5-year OS of the patients in advanced stage such as T3 (*p* = 0.791), T4 (*p* = 0.798), group of lymph node positive (*p* = 0.644). The effect of adjuvant treatment was not beneficial even in advanced stage. A study regarding effects of chemotherapy in advanced GBC is warranted.

Additionally, it can be inferred from our results that high levels of preoperative tumor markers may indicate high incidence of recurrence even in early stages of the disease. Therefore, radical surgery or chemotherapy should be considered in these cases, although further research is warranted in detail. 

Our study was limited by its single-center, retrospective nature. In addition, we failed to combine both markers in our data due to the lack of patients with elevated CEA levels. Nevertheless, we present our statistical analysis of CA19-9 and its possible use with a new cut-off value based on statistical analysis for the prognosis of GBC. This study also presents a large number of cases, much higher than those reported in previous studies. Applying this new cut off value to patients with advanced, inoperable stages of GBC may not be feasible, because our study consists of a cohort who underwent curative surgery. Further research is warranted for a proper cut-off value that is applicable for such patients.

## 5. Conclusions

In conclusion, 65 IU/mL of CA 19-9 might be considered the new cut-off value for the prognosis of GBC. Moreover, it could be an effective indicator for determining the surgical extent and the need for adjuvant treatment.

## Figures and Tables

**Figure 1 cancers-13-01089-f001:**
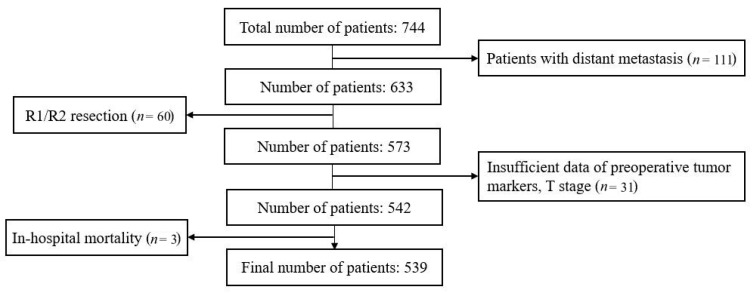
Flowchart of patient selection.

**Figure 2 cancers-13-01089-f002:**
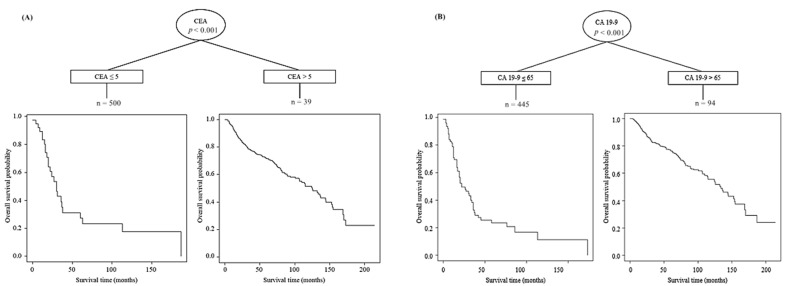
Analysis of ideal cut off value of CEA (**A**) and CA 19-9 (**B**) through C-tree method. Cut off value of CEA was 5 IU/mL and CA 19-9 was 65 IU/mL.

**Figure 3 cancers-13-01089-f003:**
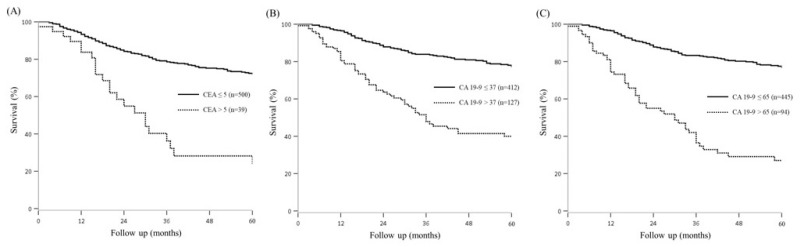
Survival graph according to CEA level and CA 19-9 level. Kaplan-Meier curves for patient following cholecystectomy 5 YOSR of the CEA ≤ 5 cases was 72.1% compared to 24.2% for the CEA > 5 cases (*p* < 0.001) (**A**). 5YOSR of the CA 19-9 ≤ 37 cases was 77.2% compared with 37.8% for the CA 19-9 > 37 cases (*p* < 0.001) (**B**). 5 YOSR of the CA 19-9 ≤ 65 cases was 76.8% compared with 24.0% for the CA 19-9 > 65 cases (*p* < 0.001) (**C**).

**Table 1 cancers-13-01089-t001:** Overall patient characteristics (*N* = 539).

Variables *n* = 539			CA 19-9 ≤ 37 *n* = 412	CA 19-9 > 37 *n* = 127	*p*-Value	CA 19-9 ≤ 65 *n* = 445	CA 19-9 > 65 *n* = 94	*p*-Value
Age (years)	65.3 (10.4)		65.5 (10.6)	64.7 (10.0)	0.072	65.4 (10.6)	64.8 (9.6)	0.121
Gender	Male	255 (47.3)	203 (49.3)	52 (40.9)	<0.001	215 (48.3)	40 (42.6)	0.007
	Female	284 (52.7)	209 (50.7)	75 (59.1)		230 (51.7)	54 (57.4)	
Total bilirubin (mg/dL)	≤1.3	362 (67.2)	282 (68.4)	80 (63.0)	0.038	306 (68.8)	56 (59.6)	0.005
	>1.3	177 (32.8)	130 (31.6)	47 (37.0)		139 (31.2)	38 (40.4)	
Operation type	Simple	172 (31.9)	142 (34.5)	30 (23.6)	0.599	152 (34.2)	20 (21.3)	0.265
	Extended	305 (56.6)	239 (58.0)	66 (52.0)		259 (58.2)	46 (48.9)	
	Others	62 (11.5)	31 (7.5)	31 (24.4)		34 (7.6)	28 (29.8)	
Differentiation	WD	187 (34.7)	163 (39.6)	24 (18.9)	0.003	170 (38.2)	17 (18.1)	0.008
	MD	196 (36.4)	128 (31.1)	68 (53.5)		145 (32.6)	51 (54.3)	
	PD	51 (9.5)	37 (9.0)	14 (11.0)		40 (9.0)	11 (11.7)	
	N/A	105 (19.5)	84 (20.4)	21 (16.5)		90 (20.2)	15 (16.0)	
T stage	T1	143 (26.5)	131 (31.8)	12 (9.4)	0.131	136 (30.6)	7 (7.4)	0.053
	T2	296 (54.9)	232 (56.3)	64 (50.4)		254 (57.1)	42 (44.7)	
	T3	94 (17.4)	45 (10.9)	49 (38.6)		51 (11.5)	43 (45.7)	
	T4	6 (1.1)	4 (1.0)	2 (1.6)		4 (0.9)	2 (2.1)	
Node metastasis	Positive	144 (26.7)	80 (19.4)	64 (50.3)	<0.001	91 (20.5)	53 (56.4)	<0.001
	Negative	395 (73.3)	332 (80.6)	63 (49.6)		354 (79.6)	41 (43.6)	
Complication	Yes	45 (8.3)	21 (5.1)	24 (18.9)	<0.001	25 (5.6)	20 (21.3)	<0.001
	No	494 (91.7)	391 (94.9)	103 (81.1)		420 (94.4)	74 (78.7)	
Chemotherapy	Yes	85 (15.8)	55 (13.3)	30 (23.6)	<0.001	60 (13.5)	25 (26.6)	<0.001
	No	454 (84.2)	357 (86.7)	97 (76.4)		385 (86.5)	69 (73.4)	
Radiotherapy	Yes	76 (14.1)	45 (10.9)	31 (24.4)	<0.001	50 (11.1)	26 (27.7)	<0.001
	No	467 (85.9)	367 (89.1)	96 (75.6)		395 (88.8)	68 (72.3)	
Recurrence	Yes	131 (24.5)	76 (18.4)	55 (43.3)	<0.001	83 (18.7)	48 (51.1)	<0.001
	No	408 (75.7)	336 (81.6)	72 (56.7)		362 (81.3)	46 (48.9)	

CEA, carcinoembryonic antigen; CA 19-9, carbohydrate antigen 19-9; WD, well differentiated; MD, moderately differentiated; PD, poorly differentiated. mean or N (SD or %).

**Table 2 cancers-13-01089-t002:** Prognostic factors for overall survival: uni and multivariated analysis.

Variable	Patients (*n* = 539)	2-Year OS (%)	5-Year OS (%)	*p*	Multivariate		
					HR	95% CI	*p*
Sex male/female	255/284	81.8/82.7	68.1/69.7	0.649			
Age (years) ≤ 60/>60	164/375	84.9/81.0	71.4/67.8	<0.001	2.245	1.562–3.226	<0.001
Total bilirubin (mg/dL) ≤ 1.3/> 1.3	362/177	85.5/76.1	72.5/62.6	0.710			
CEA (ng/mL) ≤ 5/>5	500/39	84.2/54.9	72.1/24.2	0.029	1.613	1.037–2.510	0.034
CA19-9 (IU/mL) ≤ 65/>65	445/94	87.8/55.0	76.8/24.0	<0.001	2.557	1.763–3.710	<0.001
Operation type Simple/Extended	172/305	86.5/84.7	69.8/74.4	0.004	0.567	0.399–0.804	0.001
T stage 1–2/T stage 3–4	439/100	87.5/57.8	75.8/38.0	0.015	1.752	1.162–2.641	0.007
N negative/positive	395/144	89.4/62.5	78.7/42.1	<0.001	2.341	1.668–3.287	<0.001
Complication Y/N	45/494	65.9/83.7	50.2/70.6	0.672			
Chemotherapy Y/N	85/454	67.2/85.4	51.0/72.8	0.353			
Radiation therapy Y/N	76/463	63.6/85.6	48.7/72.7	0.189			

YSR: Year-survival rate, HR: hazard ratio, CI: confidence interval, Complication: defined as over Clavien Dindo 3a.

## Data Availability

The data presented in this study are available on request from the corresponding author

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
