# Peer review of "Prognostic Value of Carcinoembryonic Antigen (CEA) and Carbohydrate Antigen 19-9 (CA 19-9) in Gallbladder Cancer; 65 IU/mL of CA 19-9 Is the New Cut-Off Value for Prognosis"

_cancers, 2021, doi:10.3390/cancers13051089_

Round 1
Reviewer 1 Report
The authors evaluated the prognostic value of CEA and CA 19-9 in gallbladder cancer, and suggested the new cut-off value of preoperative CA 19-9 level to predict the overall survival in gallbladder cancer patients, and the new cut-off value (65 IU/mL) had a stronger association with OS in comparison with conventional cut-off value of CA 19-9 (37 IU/mL).
- Tumor markers would be important not only for predicting patients’ outcome but also for early detection of recurrence which can be improve oncologic outcome by means of early application of salvage treatment. When cutoff value of tumor marker is elevated, its power to discriminate favorable group to dismal group but its potential for early detection (sensitivity) of recurrence would be lowered. What is the real clinical significance and patients benefit of increasing the cutoff value?
- What was the novelty of this study compared to previous studies about CA 19-9 of gallbladder cancer? There were several studies about tumor markers of GB cancer as below. It seems that more detailed and comprehensive discussion is needed.
- Agrawal S et al. Does CA 19-9 have prognostic relevance in gallbladder carcinoma? J Gastrointest Cancer 2018
- Yu T et al. Preoperative prediction of survival in resectable gallbadder cancer by a combined utilization of CA 19-9 and carcinoembryonic antigen. Chin Med J (Engl) 2014
- BL Strom et al. Serum CEA and CA 19-9: potential future diagnostic or screening test for gallbladder cancer? Int J Cancer 1990
- Mochizuki T et al. Efficacy of the gallbladder cancer predictive risk score based on pathological findings: a propensity score-matched analysis. Ann Surg Oncol 2018
- Wen Z et al. Elevation of Ca 19-9 and CEA is associated with a poor prognosis in patients with resectable gallbladder carcinoma. HPB (Oxford) 2017
- Liska V et al. Evaluation of tumor markers and their impact on prognosis in gallbladder, bile duct and cholangiocellular carcinomas- a pilot study. Anticancer Res 2017
- Wang YF et al. Combined detection tumor markers for diagnosis and prognosis of gallbladder cancer. World J Gastroenterol 2014
- In addition to CA 19-9, other tumor markers such as CA242 are known to be associated with GB cancer (YF Wang et al. Combined detection tumor markers for diagnosis and prognosis of gallbladder cancer. World J Gastroenterol 2014). There was no mention about other tumor markers of GB cancer in the discussion section.
- Since overall survival also can be affected by non-cancer death, cut-off value for recurrence related outcomes should be presented rather than for overall survival.
- In multivariable analysis, the adjuvant chemotherapy and radiotherapy was not associated with survival, and the authors commented in the discussion section that there have been suspicions of the effect of these adjuvant therapies in some studies. However, the effect of adjuvant treatment could not be properly evaluated because this study included considerable number of patients with early stage in terms of 26.5% of T1 and 73.3% of N0 disease.
- In previous study (Yusuke Yamamoto et al. Surgical indications for advanced gallbladder cancer considering the optimal preoperative carbohydrate antigen 19-9 cutoff value. Dig Surg 2020), the authors divided patients according to CA 19-9 level of 250, and presented groups that could be meaningful in surgical treatment in combination with clinical factors such as jaundice, major hepatectomy, and pancreticoduodenectomy. Can the present study give a clinically meaningful change in the determining of treatment strategy?
Author Response
- Reviewer A:
Comments and Suggestions for Authors
The authors evaluated the prognostic value of CEA and CA 19-9 in gallbladder cancer, and suggested the new cut-off value of preoperative CA 19-9 level to predict the overall survival in gallbladder cancer patients, and the new cut-off value (65 IU/mL) had a stronger association with OS in comparison with conventional cut-off value of CA 19-9 (37 IU/mL).
- Tumor markers would be important not only for predicting patients’ outcome but also for early detection of recurrence which can be improve oncologic outcome by means of early application of salvage treatment. When cutoff value of tumor marker is elevated, its power to discriminate favorable group to dismal group but its potential for early detection (sensitivity) of recurrence would be lowered. What is the real clinical significance and patients benefit of increasing the cutoff value?
Answer:
As you mentioned, clinical availability depends on the level of the cut-off value. , In general, when we lower the cut-off value of the tumor markers, the sensitivity for detection of malignancy may be higher, but it becomes limited for its clinical efficacy. In a similar aspect, it is very important to set a proper level of the cut-off value for predicting the prognosis including recurrence as well.
In our study, when the rate of the recurrence associated with the cut-off value for predicting 5-year OS was analyzed, the rate was 51.1% in patients with CA 19-9 levels above 65 IU/mL, compared with 24.3%that of total patients. Especially, it was found to be 28.6% for stage I, and 47.4% for stage II in patients with CA 19-9 levels above 65 IU/mL.
This content has been added in the section of results.
Therefore, even in early stage of the disease with high level of the preoperative tumor markers could indicate a high incidence of recurrence, radical surgery or chemotherapy should be considered in such cases.
This content has been added in the section of discussion.
- What was the novelty of this study compared to previous studies about CA 19-9 of gallbladder cancer? There were several studies about tumor markers of GB cancer as below. It seems that more detailed and comprehensive discussion is needed.
- Agrawal S et al. Does CA 19-9 have prognostic relevance in gallbladder carcinoma? J Gastrointest Cancer 2018
- Yu T et al. Preoperative prediction of survival in resectable gallbladder cancer by a combined utilization of CA 19-9 and carcinoembryonic antigen. Chin Med J (Engl) 2014
- BL Strom et al. Serum CEA and CA 19-9: potential future diagnostic or screening test for gallbladder cancer? Int J Cancer 1990
- Mochizuki T et al. Efficacy of the gallbladder cancer predictive risk score based on pathological findings: a propensity score-matched analysis. Ann Surg Oncol 2018
- Wen Z et al. Elevation of Ca 19-9 and CEA is associated with a poor prognosis in patients with resectable gallbladder carcinoma. HPB (Oxford) 2017
- Liska V et al. Evaluation of tumor markers and their impact on prognosis in gallbladder, bile duct and cholangiocellular carcinomas- a pilot study. Anticancer Res 2017
- Wang YF et al. Combined detection tumor markers for diagnosis and prognosis of gallbladder cancer. World J Gastroenterol 2014
Answer:
The novelty of this study compared to previous studies are as follows; Compared to other studies that used 37 IU/mL, the currently used normal upper limit of CA 19-9, we suggested a new cut off value, 65 IU/mL of CA 19-9, a value derived from statistical analysis. Additionally, it was a very important point in our study whether an adequate number was assigned to each group or not when the total cohort was divided into two groups based on 65 IU/mL. On the other hand, I think the following shortcomings are in the reports you provided.
- Agrawal S et al. Does CA 19-9 have prognostic relevance in gallbladder carcinoma?
- Number of total cases are small, cut-off value was not derived from statistical method
- Yu T et al Preoperative prediction of survival in resectable gallbladder cancer by a combined utilization of CA 19-9 and carcinoembryonic antigen.
- Number of total cases are small, cut-off value was not derived from statistical method
- BL Strom et al. Serum CEA and CA 19-9: potential future diagnostic or screening test for gallbladder cancer?
- Performed by multiple centers. cut-off value was not derived from statistical method
- Mochizuki T et al. Efficacy of the gallbladder cancer predictive risk score based on pathological findings: a propensity score-matched analysis.
- Number of total cases are too small, cut-off value was not derived from statistical method
- Wen Z et al. Elevation of Ca 19-9 and CEA is associated with a poor prognosis in patients with resectable gallbladder carcinoma.
- Cut-off value was not derived from statistical method
- Liska V et al. Evaluation of tumor markers and their impact on prognosis in gallbladder, bile duct and cholangiocellular carcinomas- a pilot study.
- Number of total cases are too small, excessively various kinds of diseases (including bile duct carcinoma, cholangiocellular carcinoma…), cut-off value was not derived from statistical method
- Wang YF et al. Combined detection tumor markers for diagnosis and prognosis of gallbladder cancer. World J Gastroenterol 2014
- excessively various kinds of diseases (including benign gallbladder disease..), cut-off value was not derived from statistical method
This contents has been added in the section of discussion.
- In addition to CA 19-9, other tumor markers such as CA242 are known to be associated with GB cancer (YF Wang et al. Combined detection tumor markers for diagnosis and prognosis of gallbladder cancer. World J Gastroenterol 2014). There was no mention about other tumor markers of GB cancer in the discussion section.
Answer:
It is very kind of you that introduce beneficial reference to us. We will add this in our section of discussion.
This contents has been added in the section of discussion.
- Since overall survival also can be affected by non-cancer death, cut-off value for recurrence related outcomes should be presented rather than for overall survival.
Answer:
It was revealed that 5-year disease-free survival rate (5-year DFS) of the patients who had a CA 19-9 level below 65 IU/mL was 73.7%, and was 23.9% in patients with CA 19-9 levels above 65 IU/mL.
This contents has been added in the section of results.
- In multivariable analysis, the adjuvant chemotherapy and radiotherapy was not associated with survival, and the authors commented in the discussion section that there have been suspicions of the effect of these adjuvant therapies in some studies. However, the effect of adjuvant treatment could not be properly evaluated because this study included considerable number of patients with early stage in terms of 26.5% of T1 and 73.3% of N0 disease.
Answer:
Subgroup analysis on the effects of chemotherapy was done in the advanced staged groups. If all T3 stage patients were divided into two groups according to chemotherapy, 47.3% was reported as 5-year OS in patients without adjuvant chemotherapy, and 45.8% was reported in group of receiving chemotherapy (p = 0.791). For T4 stage patients, it was calculated as 0% of 5-year OS for both groups (p = 0.798). When investigating lymph node positive patients, 46.2% of 5-year OS for patients without adjuvant chemotherapy and 36.1% of 5-year OS for patients with adjuvant chemotherapy were calculated (p = 0.644). For stage I patients, 85.4% and 66.7% were respectively derived as 5-year OS for patients without chemotherapy and group who received chemotherapy (p = 0.159). In stage II, 76.5% and 59.2% were respectively derived as 5-year OS for patients without chemotherapy and patients received chemotherapy (p = 0.324). In stage III patients, 47.3% of 5-year OS for group without chemotherapy and 45.8% of 5-year OS for patients received chemotherapy were calculated (p = 0.791). In stage IV, it was calculated as 0% of 5-year OS for both groups (p = 0.798). Therefore, the benefits of adjuvant chemotherapy was unclear in our study.
This contents has been added in the section of discussion.
- In previous study (Yusuke Yamamoto et al. Surgical indications for advanced gallbladder cancer considering the optimal preoperative carbohydrate antigen 19-9 cutoff value. Dig Surg 2020), the authors divided patients according to CA 19-9 level of 250, and presented groups that could be meaningful in surgical treatment in combination with clinical factors such as jaundice, major hepatectomy, and pancreticoduodenectomy. Can the present study give a clinically meaningful change in the determining of treatment strategy?
Answer:
It is obviously very applaudable to combine the level of tumor markers with other clinical factors, but our experimental period was too long to collect other various clinical data.
It is natural that higher level of tumor markers indicates higher possibility of advanced GBC. However, it is important that how many patients could be applied by suggested cut off value. In our study, patients who have over 250 IU/mL of CA 19-9 only 8.3%. In similar reason, 5 IU/mL was derived as cut off value for CEA, but the proportion of the group who have higher level of CEA than 5 IU/mL was only 7.3%, it was the reason that we couldn’t combine CA 19-9 with CEA.
This contents has been added in the section of discussion as limitation of our study.
Reviewer 2 Report
The authors present a large cohort of GBC patients in which they investigate the prognostic value of CA19-9 and CEA. They propose a new cut-off value for CA19-9 as a prognostic marker for GBC. The findings of this study are of interest to GBC clinicians. I have a few questions and suggestions to the authors to improve the quality of their manuscript.
Introduction:
- “Several studies have…for CA19-9” > please specify what the caveats of these studies were and what your study will be adding to the studies cited?
- “Furthermore, it is … and adjuvant chemotherapy” > please specify.
- Why would we need a method that can predict prognosis prior to surgery, whereas postoperative pathological analysis is considered superior by the authors? Since the majority of advanced patients is not eligible for surgery at time of diagnosis, I think this group will benefit more from a novel systemic biomarker that predicts prognosis rather than patients which received curative surgery.
Materials and Methods:
- Either remove Figure 1 or shorten the description within the main text, because with such a detailed description in the main text, Figure 1 is redundant.
Results:
- Table 1: Table header seems incorrect, since it mentions CA-19-9 twice, while CEA levels are absent. In addition, it is unclear where the cut-off value of 65 was derived from. It later appears to be determined from the C-tree method, but it is a bit confusing that these data are already presented in Table 1.
- Table 2, 2YSR and 5YSR: please specify that these concern overall survival by changing the header into: 2-year OS and 5-year OS.
Discussion:
- In general the discussion is written quite incoherent which impairs the readability. The authors should consider to improve this aspect.
- Could the authors speculate to which extent the results will be applicable for patients that have not undergone a curative resection (e.g. those with M1 disease or positive margins etc.).
- Why did the authors not investigate, for example, fibrinogen as tumor marker as well since the study of Wei et al demonstrates prognostic role for fibrinogen.
Minor/textual:
Materials and Methods:
- Please abbreviate adenocarcinoma as AC instead of ADC
- “Continuous variables were…the Student’s t-test”: please remove comma
Results:
- “With respect to T-stage…1.1% were T4” > please change into “were of stage T1… of stage T2, were of stage T3 and were of stage T4”.
- Please abbreviate 5-year overall survival rate and 5-year disease-free survival rate as 5-year OS (instead of 5YOSR) and 5-year DFS (instead of 5YDFSR), respectively.
- Figure 2: typo in rightmost panel (CA>19-9)
Discussion:
- “Consequently, 5 IU/mL and…our statistical analysis” > please rephrase sentence.
Author Response
- Reviewer B:
Comments and Suggestions for Authors
The authors present a large cohort of GBC patients in which they investigate the prognostic value of CA19-9 and CEA. They propose a new cut-off value for CA19-9 as a prognostic marker for GBC. The findings of this study are of interest to GBC clinicians. I have a few questions and suggestions to the authors to improve the quality of their manuscript.
- Introduction:
- “Several studies have…for CA19-9” > please specify what the caveats of these studies were and what your study will be adding to the studies cited?
The contents regarding this matter has been added. (Furthermore, our study focused on the prognosis as according to tumor markers because preoperatively predicted prognosis is very important to determine the treatment strategy for GBC)
- “Furthermore, it is … and adjuvant chemotherapy” > please specify.
This matter has been modified.
- Why would we need a method that can predict prognosis prior to surgery, whereas postoperative pathological analysis is considered superior by the authors? Since the majority of advanced patients is not eligible for surgery at time of diagnosis, I think this group will benefit more from a novel systemic biomarker that predicts prognosis rather than patients which received curative surgery.
Answer:
Although the postoperative pathologic finding is the most powerful prognostic factor, strategy of the operation of GBC depends on preoperative T stage which is confirmed by radiologic methods. Therefore, we have tried to find significant cut off value of tumor markers for GBC, that has been widely used in clinical field because preoperative assessment is very important.
- Materials and Methods:
Either remove Figure 1 or shorten the description within the main text, because with such a detailed description in the main text, Figure 1 is redundant.
This matter has been modified.
- Results:
- Table 1: Table header seems incorrect, since it mentions CA-19-9 twice, while CEA levels are absent. In addition, it is unclear where the cut-off value of 65 was derived from. It later appears to be determined from the C-tree method, but it is a bit confusing that these data are already presented in Table 1.
Answer:
It is natural to show the demographics of total patients prior to that of groups which is divided by 37 IU/mL and 65 IU/mL of CA 19-9. However, we wanted to reduce the space which was used to place the tables. So, these two cut-off values were shown in table header simultaneously.
- Table 2, 2YSR and 5YSR: please specify that these concern overall survival by changing the header into: 2-year OS and 5-year OS.
This matter has been modified.
- Discussion:
- In general the discussion is written quite incoherent which impairs the readability. The authors should consider to improve this aspect.
This problem has been solved by changing the composition and arrangement of some sentences.
- Could the authors speculate to which extent the results will be applicable for patients that have not undergone a curative resection? (e.g. those with M1 disease or positive margins etc.).
Answer:
For patients with inoperable status, such as group of M1 stage and R2 status, total number of patients was 129, and the, proportion of patients with CA 19-9 level below 65 IU/mL was 50.4%, and 49.6% in patients with CA 19-9 levels above 65 IU/mL. To explore 5-year OS of each group, 7.4% was derived for former group and 0% was obtained for latter group (p = 0.932).
This contents has been added in the section of discussion as limitation of our study.
When we inspected patients with R1 resection, total number was 42, and proportion of who had a CA 19-9 level below 65 IU/mL was 64.3%, and 35.7% in patients with CA 19-9 levels above 65 IU/mL. In terms of 5-year OS, 40.7% was derived for former group and 0% was obtained for latter group (p = 0.001).
- Why did the authors not investigate, for example, fibrinogen as tumor marker as well since the study of Wei et al demonstrates prognostic role for fibrinogen?
Answer:
Fibrinogen is a kind of acute-phase reactant that is synthesized in the liver and secreted into the circulation. Also, it is known that the fibrinogen levels increase in response to most forms of tissue injury, infection, or inflammation. Furthermore, in a mice study, it was reported that fibrinogen promote that lymphatic and hematogenous metastases, which led to Wei et al suggesting the combination of fibrinogen with CA 19-9 as indicator of prognosis for GBC.
However, we thought CEA and CA 19-9 had more significant prognostic values for GBC than other factors. Therefore, our study focused on demonstrating their usefulness.
This contents has been added in the section of discussion.
- Minor/textual:
- Materials and Methods:
- Please abbreviate adenocarcinoma as AC instead of ADC
This matter has been modified.
- “Continuous variables were…the Student’s t-test”: please remove comma
This matter has been modified.
- Results:
- “With respect to T-stage…1.1% were T4” > please change into “were of stageT1… of stage T2, were of stage T3 and were of stage T4”.
This matter has been modified.
- Please abbreviate 5-year overall survival rate and 5-year disease-free survival rate as 5-year OS (instead of 5YOSR) and 5-year DFS (instead of 5YDFSR), respectively.
This matter has been modified.
- Figure 2: typo in rightmost panel (CA>19-9)
This matter has been modified.
- Discussion:
“Consequently, 5 IU/mL and…our statistical analysis” > please rephrase sentence.
We have modified this problem.
Reviewer 3 Report
Job well done. But I would organize the data better, they are a bit confusingAuthor Response
- Reviewer C:
Comments and Suggestions for Authors
Job well done. But I would organize the data better, they are a bit confusing.
Thank you for your kindness. This problem has been solved by editing the contents and arrangement of some paragraphs.
Round 2
Reviewer 1 Report
The manuscript was well revised according to the comment.